# Vibrational enhancement of quadrature squeezing and phase sensitivity in resonance fluorescence

Jake Iles-Smith[1,2], Ahsan Nazir [2] & Dara P.S. McCutcheon[3]

Vibrational environments are commonly considered to be detrimental to the optical emission properties of solid-state and molecular systems, limiting their performance within quantum information protocols. Given that such environments arise naturally it is important to ask whether they can instead be turned to our advantage. Here we show that vibrational interactions can be harnessed within resonance fluorescence to generate optical states with a higher degree of quadrature squeezing than in isolated atomic systems. Considering the example of a driven quantum dot coupled to phonons, we demonstrate that it is feasible to surpass the maximum level of squeezing theoretically obtainable in an isolated atomic system and indeed come close to saturating the fundamental upper bound on squeezing from a two-level emitter. We analyse the performance of these vibrationally-enhanced squeezed states in a phase estimation protocol, finding that for the same photon flux, they can outperform the single mode squeezed vacuum state.

---

[1] Department of Physics and Astronomy, University of Sheffield, Sheffield S3 7RH, UK. [2] School of Physics and Astronomy, The University of Manchester, Oxford Road, Manchester M13 9PL, UK. [3] Quantum Engineering Technology Labs, H. H. Wills Physics Laboratory and Department of Electrical and Electronic Engineering, University of Bristol, Bristol BS8 1FD, UK. Correspondence and requests for materials should be addressed to D.P.S.M. (email: dara.mccutcheon@bristol.ac.uk)

Quadrature squeezed light has been identified as an important resource for continuous variable quantum information applications[1–6]. By reducing the variance of the electric field with respect to some phase below that of the vacuum, squeezed states can be used to increase accuracy in interferometric measurements for metrology applications[7], for secret encodings in quantum key distribution[5,8,9], and are an essential resource in continuous variable quantum computing schemes[4,6]. Experimentally, squeezed states of light can be produced by a number of different methods[10], and a wide range of classes have been explored theoretically. These include the canonical Gaussian squeezed coherent states, as well as various non-Gaussian squeezed states obtained for example via photon addition or subtraction, or constructing superpositions of Gaussian coherent states[11]. To date, the record level of squeezing has been achieved in a squeezed vacuum state using an optical parametric amplifier[12]. We note, however, that the level of squeezing is not the only consideration for applications[13].

In 1981, Walls and Zoller[14] investigated an intriguing source of non-Gaussian quadrature squeezing, which is generated in the multimode resonance fluorescence field of a driven two-level emitter (TLE), where the emitted photons are antibunched[15,16]. This was recently demonstrated experimentally using a semiconductor quantum dot platform[17], which offers the necessary high photon collection efficiency unobtainable in most atomic approaches[18–20]. The TLE scheme relies on the build-up of steady-state coherence between the ground and excited state, i.e. a state of the form $|g\rangle + c|e\rangle$ with some appreciable $c$[14]. This coherence, as well as the saturability of the emitter (which leads to antibunching) are inherited by the field, and together these properties give rise to photon statistics which amount to squeezing of a particular field quadrature. In the standard regime of atomic resonance fluorescence only a restricted set of atomic coherences (values of $c$) can be explored, and squeezing is achievable only in the weak-driving Rayleigh (equivalently Heitler) limit[19–26], with squeezing values considerably smaller than the fundamental bound for a two-level system[27–32]. Furthermore, it has yet to be explored what applications might make use of states produced in this way, and whether they offer any advantages over the more commonly studied single mode squeezed coherent states.

In this work we establish how to generate quadrature squeezed states at stronger driving above saturation, resulting in levels of squeezing that can surpass the (atomic) Walls and Zoller maximum. Our approach harnesses the vibrational environment commonly present in solid-state and molecular emitters to access states otherwise unreachable in conventional resonance fluorescence. In particular, we exploit thermalisation processes within the driving-induced dressed state basis to obtain a non-equilibrium steady state with significant coherence above saturation[22], where it would usually be strongly suppressed. Moreover, we analyse the performance of these resonance fluorescence states in a simple phase estimation protocol, finding that they are able to outperform the single mode squeezed vacuum state, and that this is only achievable with the vibrational interactions included. As a concrete example, we illustrate this behaviour through a microscopic model of a driven semiconductor quantum dot coupled to a phonon environment[33–36]. We show that off-resonant driving can be used to access levels of squeezing close to the fundamental upper bound for a two-level system, beyond those possible in the Rayleigh scattering limit below saturation (and hence any parameters for which the vibrational environment is absent). We thus identify a scenario in which vibrational processes in solid-state emitters can be used to generate squeezed states and enhance phase sensitivity in a way not possible in their absence.

## Results

**Squeezing in atomic resonance fluorescence.** Let us begin by introducing the basics of squeezing, and how it can arise in the field produced by a TLE in the standard setting without any additional vibrational environment[14]. The electric field quadrature relative to a phase reference $\varphi$ is defined as $E(t, \varphi) = e^{i\varphi}E^{(+)}(t) + e^{-i\varphi}E^{(-)}(t)$ with $E^{(+)}(t)$ the positive frequency component of the electric field, and the quadrature variance is $\Delta E(t, \varphi)^2 = \langle E(t, \varphi)^2 \rangle - \langle E(t, \varphi)\rangle^2$. The Heisenberg uncertainty relation bounds the variances in two out of phase quadratures via $\Delta E(t,\varphi)\Delta E(t,\varphi + \pi/2) \geq \frac{1}{2}|\langle[E(t,\varphi), E(t, \varphi + \pi/2)]\rangle|$. We say that a state of the field is a minimal uncertainty state when the bound is saturated, and the field is said to be squeezed if there is a quadrature that satisfies $\Delta E(t,\varphi)^2 < \frac{1}{2}|\langle[E(t,\varphi), E(t, \varphi + \pi/2)]\rangle|$. When considering emission from a TLE within the dipole approximation and the far field limit, the field may be written in the Heisenberg picture as $E^{(+)}(t) = E_0(t) - \sqrt{2\Gamma/\pi}\sigma(t)$[37,38]. The first term describes the field in the absence of the TLE, which we assume to be in the vacuum state. The second term describes the TLE emission, where the dipole operator is $\sigma = |g\rangle\langle e|$ with the ground state $|g\rangle$ and excited state $|e\rangle$, and $\Gamma$ is the spontaneous emission rate. The uncertainty relation for the field in the steady state can be written in terms of the TLE quadrature, $X(\varphi) = e^{i\varphi}\sigma + e^{-i\varphi}\sigma^\dagger$, as

$$\Delta X(\varphi)\Delta X(\varphi + \pi/2) \geq |\langle 2\sigma^\dagger\sigma - 1\rangle|, \tag{1}$$

where expectation values are taken in the long time limit. For the TLE quadrature, we have $\Delta X(\varphi)^2 = 1 - \langle X(\varphi)\rangle^2$, showing that the quadrature with the smallest fluctuations is that with the greatest expectation value, meaning that any squeezing can therefore be thought of as amplitude squeezing. Without loss of generality we can write $\langle\sigma\rangle = |\langle\sigma\rangle|e^{-i\phi}$, from which we find $\langle X(\varphi)\rangle^2 = 4|\langle\sigma\rangle|^2\cos^2(\phi - \varphi)$, and we see that the quadratures with the lowest and largest variances have $\langle X(\varphi = \phi)\rangle^2 = 4|\langle\sigma\rangle|^2$ and $\langle X(\varphi = \phi + \pi/2)\rangle^2 = 0$, respectively. The squeezing condition becomes equivalent to $:\Delta X(\phi)^2: < 0$, where we define the normally ordered quantity $:\Delta X(\phi)^2: := \Delta X(\phi)^2 - |\langle 2\sigma^\dagger\sigma - 1\rangle|$. The angle $\phi$ that defines the minimum uncertainty quadrature is the phase of the TLE dipole, which depends on the detuning of the driving laser from the transition energy, and is discussed further below.

Before we determine the conditions under which squeezing occurs, it is instructive to establish a relationship between the quadrature variance and quantities more commonly used in studies of resonance fluorescence: namely the power, the coherent scattering, and antibunching. To do so we recall that the steady-state spectrum of resonance fluorescence from a TLE is proportional to $I(\omega) = \frac{1}{\pi}\text{Re}\int_0^\infty d\tau g^{(1)}(\tau)e^{-i\omega\tau}$, with $g^{(1)}(\tau) = \langle\sigma^\dagger(\tau)\sigma\rangle$ the first-order field correlation function. The coherent contribution is separated by writing $I(\omega) = I_{\text{coh}}(\omega) + I_{\text{inc}}(\omega)$ where $I_{\text{coh}}(\omega) = g^{(1)}_{\text{coh}}\delta(\omega)$ and $I_{\text{inc}}(\omega) = \frac{1}{\pi}\text{Re}\int_0^\infty d\tau[g^{(1)}(\tau) - g^{(1)}_{\text{coh}}]e^{-i\omega\tau}$, with $g^{(1)}_{\text{coh}} = \lim_{\tau\to\infty}g^{(1)}(\tau) = |\langle\sigma\rangle|^2$[21–24]. The total radiated power can be similarly separated into coherent and incoherent contributions, giving $P = \int_{-\infty}^\infty d\omega I(\omega) = \langle\sigma^\dagger\sigma\rangle = P_{\text{coh}} + P_{\text{inc}}$ with $P_{\text{coh}} = |\langle\sigma\rangle|^2 \leq P$. We note that defined in this way, these are dimensionless power contributions satisfying $0 \leq P \leq 1$ and $0 \leq P_{\text{coh}} \leq 0.25$. In terms of these power contributions the quadrature with the minimum variance has

$$:\Delta X(\phi)^2: := 1 - |2P - 1| - 4P_{\text{coh}}, \tag{2}$$

showing that squeezing only occurs when $P_{\text{coh}}$ is appreciable, and the total power $P$ takes on values close to 0 or 1. Finally, we note that the antibunching behaviour is captured by the probability to

simultaneously detect two photons $g^{(2)}(0) = \langle (\sigma^\dagger)^2 \sigma^2 \rangle$. For a TLE $\sigma^2 = 0$, leading to $g^{(2)}(0) = 0$ regardless of the parameter regime.

For a TLE without additional vibrational interactions, the effectively flat frequency spectrum of the electromagnetic environment restricts the magnitude of the total and coherently scattered power, which in turn limits the level of squeezing. To see this, we consider a TLE described by a density operator $\rho$ driven by a coherent source with Rabi frequency $\Omega$ and laser-TLE detuning $\delta$, for which a zero temperature master equation can be written within a rotating frame and the rotating wave approximation as $\dot{\rho} = -(i/\hbar)[H_S, \rho] + \Gamma \mathcal{L}_\sigma[\rho]$. Here $H_S = \hbar \delta \sigma^\dagger \sigma + \frac{\hbar}{2}(\sigma \Omega + \sigma^\dagger \Omega^*)$, and the emission dissipator is $\mathcal{L}_\sigma[\rho] = \sigma \rho \sigma^\dagger - (1/2)(\sigma^\dagger \sigma \rho + \rho \sigma^\dagger \sigma)$, whose driving and detuning independent form is a consequence of the flat spectrum assumption. Solving the master equation in the steady state, i.e. when $\dot{\rho} = 0$, we find $P = \mathcal{S}/(2(\mathcal{S}+1))$ and $P_{coh} = P/(\mathcal{S}+1)$, leading to

$$: \Delta X(\phi)^2 := \frac{\mathcal{S}(\mathcal{S}-1)}{(\mathcal{S}+1)^2}, \tag{3}$$

where we have defined the saturation parameter $\mathcal{S} = s/(1+d)$ in terms of a dimensionless driving $s = 2(\Omega/\Gamma)^2$ and detuning $d = 4(\delta/\Gamma)^2$. The squeezing is greatest when Eq. (3) is minimised, which occurs for $\mathcal{S} = 1/3$, at which point $P = 1/8$, $P_{coh} = 3/32$, and $:\Delta X(\phi)^2 := -0.125$[14]. This represents the theoretical maximum squeezing obtainable in resonance fluorescence for a TLE undergoing spontaneous emission into an unstructured environment[14]. Furthermore, the Heisenberg uncertainty relation in Eq. (1) in this simple case becomes $\sqrt{\mathcal{S}^2 + 1} \geq 1$, which is saturated only when the TLE is undriven and $\mathcal{S} = 0$, and we therefore conclude that although the emitted field is squeezed when $\mathcal{S} = 1/3$ ($\sqrt{\mathcal{S}^2 + 1} \approx 1.05$), it is close to, but not in a minimal uncertainty state.

**Vibrational enhancement of squeezing.** As has been previously noted[27–32], the squeezed state obtained for a simple TLE as described above is not optimal, and is a consequence of the limited set of states available in this simple model. To verify this, we consider a TLE described by a completely generic density operator, i.e. one that is not necessarily a solution to the simple master equation above. Explicitly, we write

$$\rho = \frac{1}{2}(1 + l[(2\sigma^\dagger \sigma - 1)\cos\theta + X(\phi)\sin\theta]), \tag{4}$$

parameterised by a Bloch vector length $0 \leq l \leq 1$, polar angle $0 \leq \theta \leq \pi$ and dipole phase $0 \leq \phi \leq 2\pi$. We can then express the total power as $P = \langle \sigma^\dagger \sigma \rangle = (1/2)(1 + l\cos\theta)$ and the coherently scattered power as $P_{coh} = |\langle \sigma \rangle|^2 = (1/4)l^2 \sin^2\theta$. These expressions represent a less restricted set of values for the power contributions than before, and minimising Eq. (2) now gives $l = 1$, and $\theta = \pi/3$ ($P_{coh} = 3/16$, $P = 3/4$) or $\theta = 2\pi/3$ ($P_{coh} = 3/16$, $P = 1/4$), both of which result in $:\Delta X(\phi)^2 := -0.25$. This is the true maximum level of squeezing obtainable from a TLE, and is limited only by the two-level nature of the system[27–30]. Moreover, the uncertainty relation of Eq. (1) now reduces to $\sqrt{1 - l^2 \sin^2\theta} \geq |l\cos\theta|$. This is saturated for all $\theta$ when $l = 1$, for which $\rho$ describes a pure state, verifying that the states with maximum squeezing have minimum uncertainty. More generally, we see that the Bloch vector length $l$ can be used to parameterise how close a state is to minimum uncertainty.

How, then, can we obtain such a state within resonance fluorescence? We shall now show that naturally occurring vibrational interactions in, for example, solid-state and molecular systems can help to drive a TLE into such a state, resulting in a level of squeezing close to the maximum value of $:\Delta X(\phi)^2 := -0.25$, and

certainly greater than the value of $:\Delta X(\phi)^2 := -0.125$ obtainable in their absence. This can be understood qualitatively by considering equilibration of our system with respect to the additional vibrational environment. The density operator for a TLE driven according to the Hamiltonian $H_S$ defined previously, but now reaching thermal equilibrium due to some additional reservoir, can be written as the thermal state

$$\rho_{th} = \frac{e^{-\beta H_S}}{\text{tr}(e^{-\beta H_S})}, \tag{5}$$

where $\beta = 1/k_B T$ is the inverse temperature and tr denotes the trace operation. After performing the necessary exponentiation, we see that $\rho_{th}$ can be written in the general form shown in Eq. (4), with the Bloch vector parameters given by $l = l_{th} = \tanh(\hbar\beta\eta/2)$ with $\eta = \sqrt{\delta^2 + \Omega^2}$, $\theta = \arctan(\Omega/\delta)$ and $\phi = \arg(\Omega)$. As such, we see that for suitable choices of the adjustable Hamiltonian parameters $\Omega$ and $\delta$, we can satisfy the conditions above which lead to maximum squeezing. Explicitly, we find

$$: \Delta X(\phi)^2 := 1 - \frac{|\delta|}{\eta}l_{th} - \frac{|\Omega|^2}{\eta^2}l_{th}^2, \tag{6}$$

and if we choose $\delta = \pm\Omega/\sqrt{3}$, a minimal uncertainty state with the maximum squeezing of $:\Delta X(\phi)^2 := -0.25$ is achieved in the low temperature limit ($\hbar\beta\eta \to \infty$, $l_{th} \to 1$), while the quadrature displaying this squeezing can be chosen by adjusting the phase of $\Omega$. We emphasise that these states are achieved in the steady state, and as such persist for as long as the driving and temperature conditions stay fixed.

**Squeezing from a driven quantum dot.** We now explore the vibrational enhancement of squeezing in greater detail, and analyse the limitations of the simple argument given above. To do so we consider the example of a semiconductor quantum dot (QD) as a solid-state TLE with ground state $|g\rangle$ and excited state $|e\rangle$ describing a single exciton of energy $\hbar\omega_0$. The QD is driven by a semiclassical monochromatic laser with frequency $\omega_l$. Within a frame rotating with respect to the laser frequency and after applying the rotating wave approximation, the system Hamiltonian may be written as $H_S = \hbar\delta\sigma^\dagger\sigma + \frac{\hbar}{2}(\sigma\Omega + \sigma^\dagger\Omega^*)$ with $\delta = \omega_0 - \omega_l$ the detuning, as before. The QD couples to both vibrational and electromagnetic environments, with each steering the system towards competing equilibrium states. We make use of a variational master equation technique which allows regimes of strong QD-phonon coupling and laser driving to be explored within a robust formalism[22,24,39]. Full details of our model can be found in Methods and Supplementary Notes 1 and 2. The result is a master equation describing the QD excitonic degrees of freedom of the form $\dot{\rho}_V = -(i/\hbar)[H_r, \rho_V] + \mathcal{K}_{ph}[\rho_V] + \Gamma\mathcal{L}_\sigma[\rho_V]$, where $\rho_V$ is the reduced density operator of the QD in the variational frame and $H_r = \hbar\delta_r\sigma^\dagger\sigma + \frac{\hbar}{2}(\sigma\Omega_r + \sigma^\dagger\Omega_r^*)$. Coupling to phonons is accounted for in the renormalised detuning $\delta_r$ and driving $\Omega_r$, together with the phonon dissipator $\mathcal{K}_{ph}$. The form of this dissipator is given in Methods, and includes various phonon-assisted relaxation and dephasing processes. The strengths of these processes depend on the phonon spectral density, which is a measure of the electron−phonon coupling strength weighted by the phonon density of states, and takes the form $J(\omega) \propto \omega^3 \exp[-(\omega/\omega_c)^2]$ with cut-off frequency $\omega_c$ that is inversely proportional to the exciton size[24], and which sets the timescale of lattice relaxation. In the long-time limit the effect of the phonon dissipator is to tend the QD exciton towards thermal equilibrium with respect to the dressed eigenstates of $H_r$, while the spontaneous emission term $\Gamma\mathcal{L}_\sigma[\rho_V]$ acts to move the system towards the ground state.

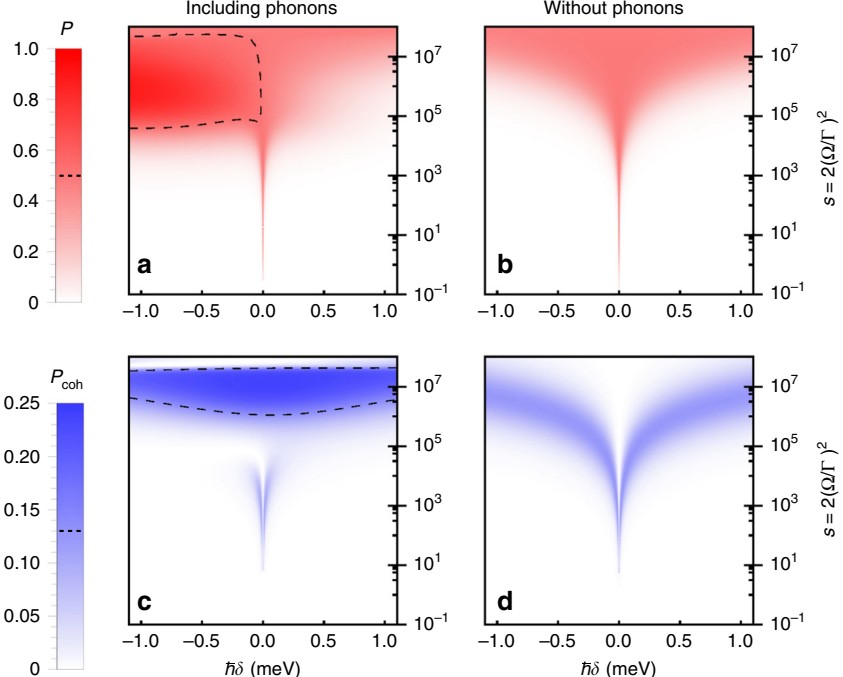

**Fig. 1** Resonance fluorescence power contributions. Normalised total emission power $P$ with (**a**) and without (**b**) phonons, and coherently scattered power $P_{coh}$ with (**c**) and without (**d**) phonons, plotted as functions of the scaled driving strength [$s = 2(\Omega/\Gamma)^2$] and detuning. Comparing the cases with and without phonons we see markedly different behaviour at large driving strengths. The dashed contour lines indicate half the maximum possible value for each power contribution from a two-level-emitter, $\frac{1}{2}P^{max} = 0.5$ and $\frac{1}{2}P_{coh}^{max} = 0.125$, neither of which are exceeded in the absence of phonons. Parameters: $\zeta = 0.027$ ps$^2$, $\omega_c = 2.2$ ps$^{-1}$, $T = 4$ K, and $1/\Gamma = 700$ ps

Having outlined our model, in Fig. 1 we show the dimensionless total power $P$ and the coherently scattered power $P_{coh}$ as functions of the dimensionless driving $s$ and the detuning $\delta$, both including (Fig. 1a and c) and excluding (Fig. 1b and d) phonons as indicated. In all cases with phonons included we define the detuning relative to a polaron shift, i.e. $\delta \rightarrow \delta - \int_0^\infty d\omega J(\omega)/\omega$. As noted for resonant driving in ref. [22], plot (c) shows that in the presence of phonons we obtain a significant coherent scattering power when driving above saturation, which we here find extends across a broad range of detunings as well. As expected, this occurs as a result of the phonons attempting to thermalise the QD exciton in the dressed state basis, which at low temperatures and strong fields leads to sustained steady-state coherence. At very high driving strengths the coherent scattering falls off, as here the generalised Rabi frequency $\sqrt{\Omega_r^2 + \delta_r^2}$ exceeds the extent of the phonon spectral density set by the cut-off $\omega_c$, leading to a regime in which the exciton and phonons are effectively decoupled[39,40].

Looking at the cases without phonons in Fig. 1b, d, we see that for all detunings the behaviour of the power contributions with increasing driving strength appears functionally the same, though with their maxima occurring at different driving strengths. This was seen in Eq. (3), which shows that the power contributions and quadrature variance depend only on the generalised saturation parameter $\mathcal{S}(\delta)$. When phonons are included, however, the situation appears more complex. On or near resonance for weak to moderate driving, the excitonic system is dominated by the spontaneous emission processes as it samples the phonon spectral density at the small generalised Rabi frequency $\sqrt{\Omega_r^2 + \delta_r^2}$. The power contributions, and hence the quadrature variance, are then similar to that of an atomic system with no phonon coupling. This can be seen in Fig. 2a, where the quadrature variance is shown as a function of the saturation

parameter on resonance, $\delta = 0$, calculated with (solid curve) and without (dotted curve) phonons included. We see that $:\Delta X(\phi)^2:$ has a minimum for $\mathcal{S}(0) = s = 1/3$, in accordance with Eq. (3). This is the regime explored experimentally in ref. [17]. Although the phonons are not playing a qualitatively significant role here, we do see that the minimum of $:\Delta X(\phi)^2:$ is slightly higher with them included. This can be attributed to the phonon sideband present in the QD emission spectrum, which acts to reduce the coherently scattered power below the level expected without phonons, even at low driving strengths[23,41,42].

Above saturation, we see that the power contributions with and without phonons markedly differ, as in this regime the phonon environment dominates over spontaneous emission due to its spectral density being sampled at larger generalised Rabi frequencies. The exciton then tends towards a thermal state as previously described, in which $P_{coh}$ can be significant, while positive and negative detunings lead to a low and high total power $P$, respectively. As anticipated, this then gives rise to two regimes with quadrature squeezing, as shown explicitly in Fig. 2b, c, where we plot the quadrature variance as a function of the generalised saturation parameter $\mathcal{S}$ for fixed detuning. The black dotted curves correspond to the approximate expression in Eq. (6), showing good agreement with the full phonon model until the driving strength becomes large enough that the decoupling regime is reached. For positive detuning we see that a level of squeezing close to the upper bound $:\Delta X(\phi)^2: = -0.25$ can be obtained, and that this is only possible when phonons are included. For the negative detuning case, as the driving increases phonons lead to thermalisation towards a state with total power $P > 0.5$ due to steady-state population inversion (see Fig. 1a)[43–45]. This means that the quadrature variance first increases at small driving, then begins to decrease with a discontinuous derivative (not shown) as $P$ passes through 0.5. The strong driving exciton–phonon decoupling regime also sets in sooner than for positive detuning.

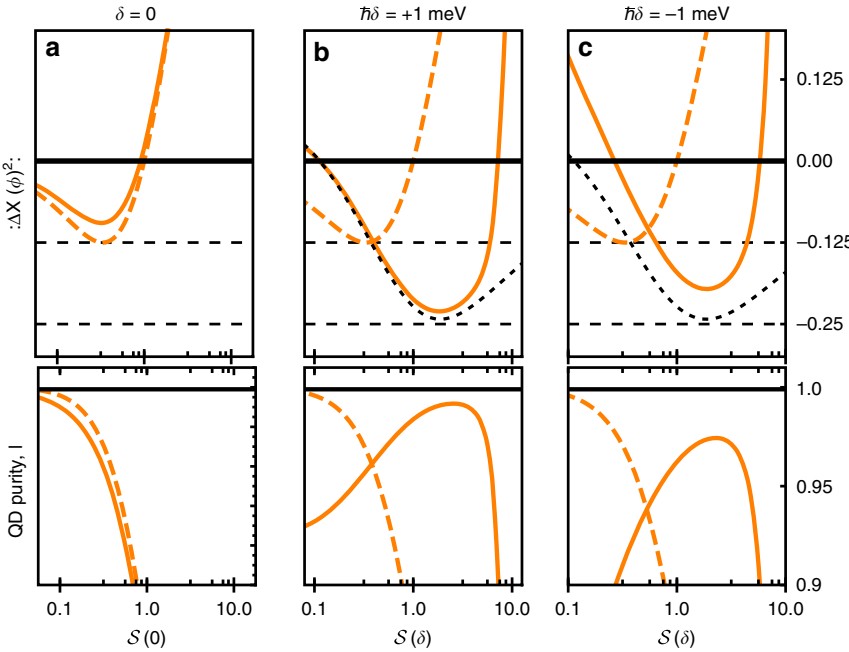

**Fig. 2** Quadrature variance and exciton purity. Normally ordered quadrature variance $:\Delta X(\phi)^2:$ on (**a**) and off (**b**, **c**) resonance. The solid curves are calculated using the full phonon theory and the dashed curves are calculated in the absence of phonons as in Eq. (3). The gridlines indicate the values $:\Delta X(\phi)^2: = -0.125$ (maximum level of squeezing obtainable in the absence of phonons) and $:\Delta X(\phi)^2: = -0.25$ (upper bound on the level of squeezing obtainable from a two-level system). The black dotted curve shows the analytic approximation in Eq. (6). The lower plots show the purity of the quantum dot excitonic state $l$, with $l = 1$ corresponding to a minimum uncertainty state of the emitted field. Parameters are as in Fig. 1. Off-resonance ($\hbar\delta = \pm1$ meV), the minimum and maximum scaled driving strengths cover the range $s \sim 10^{5.5}$–$10^{7.5}$, while in the resonant case $\mathcal{S}(0) = s$ by definition

Nevertheless, the obtained levels of squeezing still surpass those possible in the absence of the vibrational environment.

The lower plots in Fig. 2 show the corresponding purity of the QD excitonic state parameterised by its Bloch vector length $1 \geq l \geq 0$, which we use here to indicate whether the emitted field is a minimum uncertainty state (for which $l = 1$). We see that when phonons are included, the regime of greatest squeezing coincides with the regime of greatest Bloch vector length, and that the optimal values of $l = 1$ and $:\Delta X(\phi)^2: = -0.25$ are closely approached in the positive detuning case.

We note that although the driving strengths required to reach the regime of vibrationally enhanced squeezing are experimentally challenging, they are not excessively so. Rabi energies of several hundred μeV are fairly routinely achieved experimentally[46], and for the realistic parameters used in Fig. 2, the level of squeezing exceeds that possible in the absence of phonons for $\mathcal{S}(\delta) \approx 0.3$ corresponding to a Rabi energy of $\hbar\Omega \approx 0.8$ meV, while the maximum squeezing occurs in the regime $\hbar\Omega \approx 1.5$ meV. Broadly speaking, the vibrationally enhanced regime takes effect when $k_B T < \hbar\sqrt{\Omega^2 + \delta^2}$, and as such lowering the temperature below the 4 K used here would allow it to be observed at lower driving strengths and detuning values. We emphasise that the vibrationally enhanced squeezing regime occurs when phonon-induced dephasing is large, and indeed dominates over spontaneous emission. As such, our results are robust against any additional pure-dephasing processes that are weak or moderate compared to the homogeneous linewidth of the exciton, as shown explicitly in Supplementary Note 4.

**Resonance fluorescence and phase estimation**. Having shown how a vibrational environment is able to lead to enhanced levels of quadrature squeezing in resonance fluorescence, a natural question to ask is whether the light produced is useful in applications. To address this question, we analyse the canonical phase

estimation protocol illustrated in Fig. 3. It consists of an unbalanced Mach−Zehnder interferometer with path difference parameterised by $\Theta$, and into which is inserted a pure coherent state $|\alpha\rangle$ in one arm and a general state $\rho_{\rm in}$ in the other arm. We are interested in the greatest accuracy with which $\Theta$ can be estimated. We consider perhaps the simplest measurement that can be used to construct an estimator for $\Theta$, which is the expectation value of the difference in photon numbers at the two outputs, $\langle N_- \rangle$, where $N_\pm = N_1 \pm N_2$. Our accuracy figure of merit is then the variance in $N_-$, $\Delta N_-^2 = \langle N_-^2 \rangle - \langle N_- \rangle^2$, normalised by the total photon flux $\langle N_+ \rangle$, which we label $\mathcal{F} = \Delta N_-^2 / \langle N_+ \rangle$. Defined in this way, if a second coherent state is input into the first (upper) arm our figure of merit gives $\mathcal{F} = 1$.

Considering first a squeezed single mode vacuum input state, we recover the seminal result of Caves[1]. As detailed in Methods, using $\rho_{\rm in} = |\xi\rangle\langle\xi|$ with $|\xi\rangle = S(\xi)|{\rm vac}\rangle$ and $S(\xi) = \exp[(\xi a^2 - \xi^* a^{\dagger 2})/2]$, we find $\langle N_+ \rangle = P_\xi + P_\alpha$ with $P_\xi = \sinh^2 |\xi|$ and $P_\alpha = |\alpha|^2$, while the variance in the difference in photon numbers is minimised for $\Theta = \pi/2$, at which point

$$\Delta N_-^2 = P_\xi + P_\alpha e^{-2|\xi|}, \tag{7}$$

showing that for $|\xi| > 0$ the variance is reduced below that of a coherent state input with the same power[1,13]. For the squeezed vacuum we find $\mathcal{F}$ is minimised for $|\xi| = (1/2)\ln[1 + 2\sqrt{P_\alpha}]$ at which point $\mathcal{F} = 1/(1 + \sqrt{P_\alpha})$.

For the resonance fluorescence field as the input state, for the same $\Theta$ the variance in $N_-$ is minimised when the phase of the coherent state amplitude $\alpha$ is equal to the dipole phase $\phi$, and is now given by

$$\Delta N_-^2 = P + P_\alpha(1 - 4P_{\rm coh}), \tag{8}$$

with $P$ and $P_{\rm coh}$ the total and coherently scattered dimensionless powers as before. Comparing Eqs. (7) and (8), we can immediately see that if $P_{\rm coh}$ can reach values closest to its

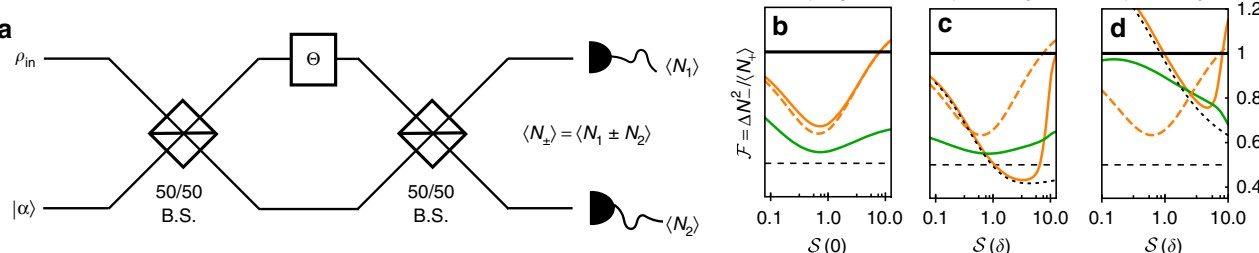

**Fig. 3** Phase estimation with vibrationally enhanced resonance fluorescence fields. **a** Phase estimation setup, showing a test input state $\rho_{in}$ in the first (upper) mode and a coherent state input $|\alpha\rangle$ in the second (lower) mode of a Mach−Zehnder interferometer consisting of two 50/50 beamsplitters separating and recombining the two arms with path length difference parameterised by $\Theta$. When the first mode is illuminated with resonance fluorescence light, the normalised phase sensitivity for $\Theta = \pi/2$ is shown for resonant (**b**) and off-resonant driving (**c**, **d**), modelled with (solid, orange) and without (dashed, orange) coupling to phonons, and also calculated using the approximate QD steady-state in Eq. (5). Shown for reference is the normalised phase sensitivity for a squeezed vacuum input state in mode one (solid, green), with the squeezing parameter chosen such that the expectation value of the photon number is equal to that in the resonance fluorescence case. The thick black line at $\mathcal{F} = 1$ is the phase sensitivity for a coherent state input in the first (upper) interferometer arm, while the dashed black line at $\mathcal{F} = 0.5$ is the global minimum for the squeezed vacuum input. Parameters are as in Fig. 1 with $|\alpha|^2 = 1$

maximum of 0.25, the second term in Eq. (8) will vanish, something that is only possible in the limit of infinite squeezing in the squeezed vacuum case.

Our figure of merit is shown in Fig. 3 as a function of driving strength, for a QD driven on (Fig. 3b) and off-resonance (Fig. 3c and d) with (solid, orange curves) and without (dashed, orange) phonons, where we take $P_\alpha = 1$. Shown also in green is the same figure of merit for the squeezed single mode vacuum, with the squeezing parameter $\xi$ chosen such that $P_\xi = P$, ensuring that the total photon flux is equal in both cases. The black dashed curve is calculated using the approximate QD steady state in Eq. (5), showing good agreement with the full phonon model in the phonon enhanced regimes.

Looking first at the resonant case (b), we see that the figure of merit at low driving strengths is minimised by the squeezed vacuum input state, implying that for a fixed photon flux, it is this state which gives the greatest sensitivity to the phase $\Theta$. Hence, the resonance fluorescence field in the standard squeezing regime does not outperform the squeezed vacuum. Off-resonance and for positive detuning, however, we see that the resonance fluorescence field can outperform the squeezed vacuum state, as in this regime $P_{coh} \approx 0.25$ when phonons are included (cf. Fig. 1). Moreover, in this regime our figure of merit $\mathcal{F}$ reaches values below 0.5, which is the minimum possible value for the squeezed vacuum for any parameters when $P_\alpha = 1$. In the negative detuning case, although $P_{coh}$ is again close to 0.25, the total power $P$ is now large, meaning the first term in Eq. (8) is significant. The competition between these two terms together with the phonon-induced population inversion gives rise to the complicated behaviour seen in Fig. 3d. Interestingly, here we see that the resonance fluorescence field without phonons can actually outperform the squeezed vacuum, although the overall squeezed vacuum minimum of $\mathcal{F} = 0.5$ is only beaten when including phonons in the positive detuning case.

## Discussion

We have shown that interactions between a TLE and a vibrational environment can be harnessed to produce a source of quadrature squeezed light with levels of squeezing that would otherwise be impossible within resonance fluorescence. In fact, the obtainable squeezing can reach values very close to the fundamental bound for a two-level system. We have illustrated our findings with an explicit example of a QD coupled to phonons, which provides a feasible experimental platform to engineer such squeezed states of light.

We have also explored how these resonance fluorescence states perform in a phase estimation protocol. Without vibrational interactions, although the reduced fluctuations in resonance fluorescence states can provide an advantage over coherent state inputs, the single mode squeezed vacuum offers the overall best phase sensitivity when minimised over the squeezing magnitude. With vibrational interactions included, however, the resonance fluorescence state can outperform the squeezed vacuum, even when the latter is operating in its optimal regime, suggesting that resonance fluorescence fields could provide a useful resource in phase estimation applications. It is interesting to compare these findings with the Fisher information analysis of ref. [13], which states that for the setup shown in Fig. 3, the optimal pure single mode input state over all phase estimators is the squeezed vacuum. Our findings therefore suggest that for more general multimode input states the squeezed vacuum ceases to be optimal, or that estimators beyond the difference in output photon numbers must be considered. As such, our results not only illustrate how vibrational environments can give rise to enhanced quadrature squeezing in resonance fluorescence, but also motivate future studies analysing the performance of generalised non-Gaussian multimode states in interferometry.

## Methods

**Quantum dot master equation.** The QD couples to both vibrational and electromagnetic environments, which results in the total Hamiltonian $H = H_S + H_{S-ph} + H_{S-em} + H_B$. At low temperatures the electron−phonon interaction is dominated by a linear displacement coupling with Hamiltonian $H_{S-ph} = \sigma^\dagger\sigma \sum_k \hbar g_k(b_k^\dagger + b_k)$, where $b_k (b_k^\dagger)$ is the annihilation (creation) operator of phonon mode $k$ with frequency $\omega_k$ and coupling strength $g_k$[24,47–49]. The electron−phonon coupling is characterised by the spectral density $J(\omega) = \sum_k |g_k|^2\delta(\omega - \omega_k) = \zeta\omega^3\exp[-\omega^2/\omega_c^2]$[24], with coupling strength $\zeta$ and cut-off frequency $\omega_c$. Coupling to the electromagnetic field in the dipole and rotating wave approximations takes the form $H_{S-em} = \hbar\sum_m h_m\sigma^\dagger a_m e^{i\nu_m t} + \text{h.c.}$, where $a_m (a_m^\dagger)$ is the annihilation (creation) operator for mode $m$ of the field, with frequency $\nu_m$ and coupling strength $h_m$. We assume the spectral density of the optical field varies slowly over the relevant energy scales of the system, allowing us to use the flat function $\sum_m |h_m|^2\delta(\nu - \nu_m) \approx 2\Gamma/\pi$, where $\Gamma$ is again the spontaneous emission rate. Free evolution of the environments is described by $H_B = \hbar\sum_k \omega_k b_k^\dagger b_k + \hbar\sum_m \nu_m a_m^\dagger a_m$.

To account for the electron−phonon coupling we make use of the variational polaron transformation[22,24,39] defined by the unitary $\mathcal{U} = |g\rangle\langle g| + |e\rangle\langle e|B_+$, where $B_\pm = \exp[\pm\sum_k f_k(b_k^\dagger - b_k)/\omega_k]$. This leads to a QD state-dependent displacement of the phonon environment, where the $f_k$ are chosen to minimise the Feynman−Bogoliubov bound on the free energy, defining an optimised basis in which perturbation theory can then be applied. In the variational polaron frame we derive a second-order Born−Markov master equation, which is valid for both strong and weak exciton−phonon coupling, as well as from weak to strong laser driving strengths. For real $\Omega$, this may be written compactly as

$\dot{\rho}_{\mathrm{V}} = -(i/\hbar)[H_r, \rho_{\mathrm{V}}] + \mathcal{K}_{\mathrm{ph}}[\rho_{\mathrm{V}}] + \Gamma\mathcal{L}_\sigma[\rho_{\mathrm{V}}]$, where $\rho_{\mathrm{V}}$ is the reduced density operator of the QD in the variational frame and the renormalised system Hamiltonian is $H_r = \hbar\delta_r\sigma^\dagger\sigma + (\hbar\Omega_r/2)(\sigma^\dagger + \sigma)$, with $\delta_r$ and $\Omega_r$ defined below. The phonon dissipator is

$$\mathcal{K}_{\mathrm{ph}}[\rho_{\mathrm{V}}] = [\widehat{\mathcal{Z}}_{zz}\rho_{\mathrm{V}}, \sigma^\dagger\sigma] - \tfrac{\Omega^2}{4}([\sigma_x, \widehat{\mathcal{X}}_{xx}\rho_{\mathrm{V}}] + [\sigma_y, \widehat{\mathcal{Y}}_{yy}\rho_{\mathrm{V}}])$$
$$+ \tfrac{i\Omega}{2}([\sigma_y, \widehat{\mathcal{Z}}_{yz}\rho_{\mathrm{V}}] - [\sigma^\dagger\sigma, \widehat{\mathcal{Y}}_{yz}\rho_{\mathrm{V}}]) + \text{h.c.} \quad (9)$$

where the rates are contained in the system operators

$$\widehat{\mathcal{X}}_{\alpha\beta} = \sum_{jk} \sigma_x^{jk} \int_0^\infty d\tau e^{i\lambda_{jk}\tau} \Lambda_{\alpha\beta}(\tau)|\psi_j\rangle\langle\psi_k|, \quad (10)$$

$$\widehat{\mathcal{Y}}_{\alpha\beta} = i\sum_{jk} \sigma_\gamma^{jk} \int_0^\infty d\tau e^{i\lambda_{jk}\tau} \Lambda_{\alpha\beta}(\tau)|\psi_j\rangle\langle\psi_k|, \quad (11)$$

$$\widehat{\mathcal{Z}}_{\alpha\beta} = i\sum_{jk} (1 + \sigma_z^{jk}) \int_0^\infty d\tau e^{i\lambda_{jk}\tau} \Lambda_{\alpha\beta}(\tau)|\psi_j\rangle\langle\psi_k|, \quad (12)$$

which are written in terms of the phonon correlation functions $\Lambda_{xx}(\tau) = (B^2/2)$ $(e^{\kappa(\tau)} + e^{-\kappa(\tau)} - 2)$, $\Lambda_{yy}(\tau) = (B^2/2)(e^{\kappa(\tau)} - e^{-\kappa(\tau)})$ and $\Lambda_{zz}(\tau) = \int_0^\infty d\omega J(\omega)$ $(1 - F(\omega))^2 C_\parallel(\tau, \omega)$, with $\kappa(\tau) = \int_0^\infty d\omega J(\omega)F(\omega)^2\omega^{-2}C_\parallel(\tau, \omega)$ and

$$C_\parallel(\tau, \omega) = \coth\left(\frac{\hbar\beta\omega}{2}\right)\cos(\omega\tau) - i\sin(\omega\tau), \quad (13)$$

while $\Lambda_{yz}(\tau) = -2B\int_0^\infty d\omega J(\omega)\omega^{-1}F(\omega)(1 - F(\omega))C_\perp(\tau, \omega)$ with

$$C_\perp(\tau, \omega) = \coth\left(\frac{\hbar\beta\omega}{2}\right)\sin(\omega\tau) + i\cos(\omega\tau). \quad (14)$$

The eigenstates of the renormalised system Hamiltonian satisfy $H_r|\psi_j\rangle = \psi_j|\psi_j\rangle$, and we have defined $\hbar\lambda_{jk} = \psi_j - \psi_k$ and $\sigma_\alpha^{jk} = \langle\psi_j|\sigma_\alpha|\psi_k\rangle$. In the above the displacement operator thermal average is $B = \langle B_\pm\rangle = \exp[-\kappa(0)/2]$ and $F(\omega)$ is the variationally determined mode displacement defined in Supplementary Note 1. The renormalised detuning and Rabi frequency are $\delta_r = \delta + \int_0^\infty d\omega J(\omega)\omega^{-1}F(\omega)(F(\omega) - 2)$ and $\Omega_r = \Omega B$.

We note that in the variational frame the dipole operator carries a displacement operator, such that $\sigma \to B_-\sigma$. This displacement operator leads to a phonon sideband, a consequence of non-Markovian lattice relaxation during the emission process[23,41,42,50]. Including this effect, the field emitted by the QD becomes $E^{(+)}(t) = E_0(t) - \sqrt{2\Gamma/\pi}B_-(t)\sigma(t)$, and we obtain a modification to the quadrature variance, which becomes (cf. Eq. (2)) $:\Delta X(\phi)^2 := 1 - |2P - 1| - 4B^2 P_{\mathrm{coh}}$. Similarly, Eq. (8) becomes $\Delta N_-^2 = P + P_\alpha(1 - 4B^2 P_{\mathrm{coh}})$.

**Phase estimation protocol.** The phase estimation setup we consider is shown and described in Fig. 3, and relies on an estimator based on the difference in intensities at the two outputs, which has corresponding dimensionless operator $N_- = N_1 - N_2$. As such we seek to analyse the variance in this quantity, $\Delta N_- = \langle N_-^2\rangle - \langle N_-\rangle^2$. To proceed we write the output number operators in the steady state as $N_1 = E_1^{(-)}E_1^{(+)}$ with $E_1^{(+)}$ the positive frequency component of the electric field operator in arm 1, and similarly for arm 2. Working in the Heisenberg picture and neglecting retardation effects, we can relate these operators at the outputs to those at the inputs using

$$\begin{pmatrix} E_1^{(+)} \\ E_2^{(+)} \end{pmatrix} = \frac{1}{2}\begin{pmatrix} E_-^{(+)}(0) + E_+^{(+)}(\Theta) \\ iE_-^{(+)}(0) - iE_+^{(+)}(\Theta) \end{pmatrix}, \quad (15)$$

where $E_\pm^{(+)}(\Theta) = E_{\mathrm{in}}^{(+)}(\Theta) \pm iE_\alpha^{(+)}(\Theta)$, with $E_{\mathrm{in}}^{(+)}(\Theta)$ the positive frequency component of the electric field at the first input arm illuminated by the state $\rho_{\mathrm{in}}$, propagated by the path difference parameter angle $\Theta$, while $E_\alpha^{(+)}(\Theta)$ is the corresponding quantity at the second input arm illuminated by the coherent state $|\alpha\rangle$. Using these relations we straightforwardly find $N_- = (1/2)[E_-^{(-)}(0)E_+^{(+)}(\Theta) + E_+^{(-)}(\Theta)E_-^{(+)}(0)]$, and where at this stage we have made no assumptions about the state of the field or the path length difference $\Theta$.

The second arm is illuminated by a single mode coherent state. As such, when expectation values are taken, all other modes do not contribute, and neglecting constants we can therefore write the dimensionless electric field operator as $E_\alpha^{(+)}(\Theta) = E_\alpha^{(+)}(0)e^{-i\Theta} = a_2 e^{-i\Theta}$. A similar argument holds when the first arm is illuminated by the single mode squeezed vacuum. In the resonance fluorescence case the situation is more subtle. Omitting the free field (vacuum) contribution, and again dropping constants, we have $E_{\mathrm{in}}^{(+)}(\Theta) = \sigma_{\mathrm{L}}(\tau_\Theta)$, where the subscript signifies a 'lab'-frame (non-rotating) operator and $\tau_\Theta$ is the time delay corresponding to the phase shift $\Theta$. In the multimode resonance fluorescence case it is not obvious that a time delay gives rise to simple phase factor on the electric field operators. However, since we work in a rotating frame, we can write $\sigma_{\mathrm{L}}(\tau_\Theta) = \sigma(\tau_\Theta)e^{-i\tau_\Theta\omega_1}$, where $\sigma$ is the rotating frame dipole operator and $\omega_1$ is the driving laser frequency. If we are interested in delays $\tau_\Theta$ that give rise to phase shifts

$\Theta = \tau_\Theta\omega_1 \approx \pi$, this corresponds to $\tau_\Theta \sim$ fs for $\omega_1 \sim 1$ eV. Over these very short times the rotating frame operator $\sigma$ does not significantly change, and we can write $E_{\mathrm{in}}^{(+)}(\Theta) = E_{\mathrm{in}}^{(+)}(0)e^{-i\Theta} = \sigma(0)e^{-i\Theta}$.

Putting these results together, in the case of the squeezed vacuum input, for which we have $\rho_{\mathrm{in}} = |\xi\rangle\langle\xi|$ with $|\xi\rangle = exp[(\xi a^2 - \xi^* a^{\dagger 2})/2]|\mathrm{vac}\rangle$, we find

$$\langle N_-\rangle = \cos\Theta(P_\xi - P_\alpha), \quad (16)$$

with $P_\xi = \sinh^2|\xi|$ the (dimensionless) power in the squeezed vacuum state, and $P_\alpha = |\alpha|^2$ the power in the coherent state. The variance in the difference in photon number for this input state is

$$\Delta N_-^2 = P_\alpha + P_\xi(P_\xi + 3/2) + \cos(2\Theta)P_\xi(P_\xi + 1/2)$$
$$+ (1 - \cos(2\Theta))P_\alpha(P_\xi + \sqrt{P_\xi(P_\xi + 1)}\cos\Delta\phi), \quad (17)$$

where we have written $\alpha = |\alpha|e^{i\phi_\alpha}$, $\xi = |\xi|e^{-i\phi_\xi}$, and $\Delta\phi = 2\phi_\alpha - \phi_\xi$. This expression is minimised for $\Theta = \pi/2$ and $\Delta\phi = \pi$, at which point we have

$$\Delta N_-^2 = P_\xi + P_\alpha(1 + 2P_\xi - 2\sqrt{P_\xi(P_\xi + 1)}),$$
$$= P_\xi + P_\alpha e^{-2|\xi|}, \quad (18)$$

which is known previously from the work of Caves[1].

In the resonance fluorescence case we use $\Theta = \pi/2$ to facilitate a fair comparison, and now find

$$\Delta N_-^2 = P + P_\alpha(1 - 4P_{\mathrm{coh}}\cos^2\Delta\phi), \quad (19)$$

where now $P = \langle\sigma^\dagger\sigma\rangle$ is the (dimensionless) power in the resonance fluorescence field, and we have written $\langle\sigma\rangle = |\langle\sigma\rangle|e^{-i\phi} = \sqrt{P_{\mathrm{coh}}}e^{-i\phi}$ with $\phi$ the dipole phase in the steady state and $\Delta\phi = \phi_\alpha - \phi$. In this case the minimum corresponds to $\Delta\phi = 0$, and with this substitution we arrive at Eq. (8).

## Data availability
The data that support the findings of this study are available from the corresponding author upon reasonable request.

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

## Acknowledgements

The authors wish to thank Alistair Brash, Pieter Kok and Jonathan Matthews for useful discussions. D.P.S.M. acknowledges funding from the European Union's Horizon 2020 research and innovation programme under the Marie Skłodowska-Curie grant agreement no. 703193, and the EPSRC, grant no. EP/L024020/1. A.N. is supported by the EPSRC, grant no. EP/N008154/1, and J.I.-S. acknowledges support from the Royal Commission for the Exhibition of 1851.

## Author contributions

J.I.-S. derived the underlying formalism and master equation, with input from all authors. D.P.S.M. derived the main results, with input from all authors. J.I.-S., A.N., and D.P.S.M. all contributed towards analysis, discussions and preparation of the manuscript.

## Additional information

**Competing interests:** The authors declare no competing interests.

