## [Peer Review File · Nature Communications]

Reviewers' comments:

Reviewer #1 (Remarks to the Author):

The authors present a theoretical treatment of a two-level system in the presence of phonons, and analyze the light emitted by this system to see whether there is a reduction of one observable of the field compared to another observable, such as the real and imaginary parts of the electric field scattered by the system. I find the paper very interesting, especially that solid state emitters have some advantage over atomic systems in this application. While this is clearly exciting, I wonder if (a) the motivation for achieving this maximum level of squeezing is well justified/explained and (b) whether this level of squeezing could actually be seen under real experimental conditions. I also have the following questions about the manuscript:

1. I find the introduction to be rather 'generic' with regards to squeezing. There are many ways to observe squeezing, and many variables which can be considered to be 'squeezed'. I think some more text should be given over to the level of squeezing observed in different situations (ie photon addition/subtraction, parametric oscillators, etc) and their inherent limitations.

2. The squeezing here arises through the fact that the driven two-level system can be parameterised by the Pauli matrices, and the expectation values of these can be related by commutation relations. The authors use this to define $\Delta X(\varphi)^2$, but it is not completely clear to me what quadrature is being considered as squeezed when using this? Is this phase being squeezed with respect to amplitude? Some more clarity on what property of the field is squeezed would be really useful.

3. When the authors move from the simple atomic case, to the case with the phonon environment, they say they introduce a 'completely generic density operator' – what do they mean by this, and how does this differ from the atomic case?

4. When phonons are introduced, they only arise in a phonon side-band to the main Lorentzian peak of the two level system of width Γ – is there no broadening of the line due to these phonon interactions, often termed 'pure dephasing'? This would also affect the saturation parameter s .

5. In the plots of Fig. 1, the authors take the saturation parameter to a level of 10^7 . Is this feasible for a solid state two level system?

6. In Fig. 2 the authors only take the detuning modified saturation parameter S from 0.1 to 10, and the maximum squeezing is around $S = 2$ (the axes would also be less confusing if labelled S and with log ticks on the lower axis of the box). As these are all at a fixed detuning would it not be better to stick to using the saturation parameter s , to allow for easy comparison between figures? Also, where on Fig. 1 corresponds to these plot parameters?

7. The Wigner functions have clear negativity, which is very interesting. The negativity (not needed for a squeezed state) is much larger than is usually observed for a Fock state of one photon. Is there a limit to the negativity that can be achieved? Could the authors comment on the nonclassicality of their states based on the negativity?

8. Finally, is the maximum level of squeezing observed useful? Can it be harnessed for any of the applications mentioned in the first paragraph of the manuscript such as QIP applications, sensitive measurements, or QKD systems?

Reviewer #2 (Remarks to the Author):

The authors discuss the problem of generating stronger squeezing in the resonance fluorescence of a two-level atom than that predicted in 1981 by Walls and Zoller. They comment that the bound predicted by Walls and Zoller is not the largest possible value to which one of the quadrature

component of the fluorescence field could be reduced. In fact, the lowest possible value is twice of that predicted by Walls and Zoller. Next, the authors show that the lowest possible value can be achieved in the case when the atom interacts with a specific bath, a phonon bath.

The authors use the terminology "Thermal enhancement ...". Of course, phonons are due to thermal oscillations but in fact the way the phonon bath couples to the atom is different than the ordinary thermal field. It is clearly seen in the Hamiltonian (1) of the supplementary material. Therefore, the title of the paper should be changed from "Thermal enhancement ..." to "Phonon enhancement....".

Also the fact that the Walls and Zoller bound can be beaten in the resonance fluorescence of a two-level atom has been known for a very long time. There are papers one by Ficek et al. JOSA B1, 882 (1984) in which it is shown that the Walls and Zoller limit can be beaten in the transient regime of the resonance fluorescence where squeezing twice of that predicted by Walls and Zoller can be achieved in short times. A more general analysis of that prediction was made in 1987 by Aravind, JOSA B4, 1847 (1987).

The fact that a phonon bath can enhance squeezing is interesting, but I am not convinced how useful it could be for practical applications. The interest is in the generation of squeezed light with significantly reduced fluctuations, I don't see how the fluorescence field with squeezed fluctuations could be applied in practice. Any idea how the fluorescence field could be applied as a source of squeezed light?

Reviewer #3 (Remarks to the Author):

Dear Editor,

please find below my report for the manuscript "Thermal enhancement of quadrature squeezing in resonance fluorescence" by Dr. Iles-Smith et.al

This manuscript illustrates how thermal interactions can be used within resonance fluorescence to generate single photons with a higher quadrature squeezing than in isolated atomic systems. More specifically they show they can generate a bright source of single photons with a level of quadrature squeezing that could surpass the traditional atomic scheme. This is an interesting and counter intuitive idea. However there are a number of serious issues that need to be addressed before publication could be considered.

The authors also need to consider the following

- first and foremost is the introduction does not really motivate what the paper is really about. The first paragraph is about application of squeezed states - generally with many photons. The scheme presented here (at least what the abstract says) is for single photons. What are the applications for squeezed single photons.
- In the introduction it is stated that squeezed states "provide the necessary non- classicality for quantum computing schemes". This is not really true. For CV computation one requires at least a third order nonlinearity. Squeezing is second order.
- the choice of $\hbar=1$ is confusing especially when the authors consider realistic parameters later in the paper. One should work in SI units and so I recommend the \hbar s are explicitly put back into the paper. One can see this when $E^t(t)=E_0(t)-\text{Sqrt}[2 \text{Gamma}/\text{Pi}] \sigma_-$. E_0 seems from the definition below to be dimensionless while the second terms is $s^{(-1/2)}$?. Notation wise it would be better to use σ_{\pm} rather than σ & σ^\dagger . Further the authors themselves refer to ω_l as both an energy and a frequency.
- Next what is $|X\rangle$ being used for excited state rather than the more traditional $|e\rangle$?
- On page 2 of the manuscript it is written "we see that the quadrature with the lowest variance is that with $\phi=\psi$ ". What happens if $\phi=\psi+\pi$?
- Next (as seen in Walls and Milburn), the spectrum is normally defined from $-\infty$ to ∞ . What is the reason to define from 0 to ∞ in this case?
- It would be useful after eqn (1) to also present the anti squeezed quadrature for completeness. This way we can check that the squeezed state generated is still is a minimum uncertainty state.
- When the master equation is first introduced on page 2, it is mentioned that "... TLE without

additional thermal interactions". The author need to be more explicit that they are working at zero temperature for the master equation presented.

- With almost any quantum dot system (and in fact solid state system), dephasing is a critical effect and generally more limiting than T_1 . What is its effect here? It probably needs to be included.
- Using the symbol S for saturation parameters is problematic as $S(\omega)$ is also used for the spectrum
- The first paragraph in the second column of page 2 begins "To verify this we consider a TLE described by a completely generic density operator". It would be useful to have a little more explanation on how the results are derived from it.
- Next when the thermal reservoir is added, are the steady state just obtained by solving the appropriate master for non zero temperature. Why is it mentioned this is an additional reservoir? Above eqn (3) it is mentioned that ρ_{th} is the TLE thermal state. If it is a thermal state by standard definitions, it should not be squeezed? Please explain
- Next around eqn (3) please show the variance of the other quadrature as well. What is the effect of the thermal noise on it. Do we still have a minimum uncertainty state? Below eqn (3) is stated that the maximum amount of squeezing is " $\Delta X(\phi)^2 = 0.25$ is achieved in the low temperature limit". This is a positive value so is it really squeezed?
- The model used here is showing not spatial character for the phonon bath but we have a 3D system. Generally when there is not spatial confinement, we need the e^{ikx} term. However as there is no cavity etc, this may be important. Please comment on it?
- The electron-phonon interaction is given by the Hamiltonian H_{PH} . However this makes it look like a coherent interaction? Is it? The g_k will vary over time in terms of both amplitude and phase. Should we not have written the last part as $g_k b_k^\dagger + g_k^* b_k$?
- The dissipator K_{ph} is not really described in the main text of the paper, nor what form it has. It is in the supp material but a description needs to be in the main text.
- When the Wigner function is described I have a problem with it. The emission of the light field is not into a single mode and in fact is radiated into free space (3D). How is this accounted for? Next it is well known that the Wigner function for the state of an atom can be negative, even if it is in its ground or excited state.
- I am also confused that the Wigner function for the squeezed state is negative. For optical fields it is strictly positive as it comes from a Gaussian state. A significant discussion needs to be made around this point.
- Finally to show that the states generated are useful, how about illustrating their application in a metrology protocol.
- Now as a more minor comments, the format of this paper is not that used for Nature Communications.

To summarize, I believe this is an interesting article, but it is not clear to me currently whether it would be of interest to the wider community. The shortened nature of the manuscript means important details are missing (which could be included given the non letter format of Nat Comm).

Vibrational enhancement of quadrature squeezing and phase sensitivity in resonance fluorescence (NCOMMS-18-21112)

Response to Reviewer #1 (Reviewer's comment in italics)

We thank the referee for their careful reading of our manuscript, and for their positive comments and suggestions. We have made a number of changes to the manuscript to address the points raised. Below we give a detailed response to each point raised in turn.

The referee gives a clear summary of our work, before stating:

1. I find the introduction to be rather generic with regards to squeezing. There are many ways to observe squeezing, and many variables which can be considered to be squeezed. I think some more text should be given over to the level of squeezing observed in different situations (ie photon addition/subtraction, parametric oscillators, etc) and their inherent limitations.

Based on the referee's comments here (and those of the other referees) we have changed the introduction of the manuscript to better put our results in context. In particular, we now place a greater emphasis on the different classes of squeezed states than can arise.

2. The squeezing here arises through the fact that the driven two-level system can be parameterised by the Pauli matrices, and the expectation values of these can be related by commutation relations. The authors use this to define : $\Delta X(\varphi)^2$: but it is not completely clear to me what quadrature is being considered as squeezed when using this? Is this phase being squeezed with respect to amplitude? Some more clarity on what property of the field is squeezed would be really useful.

We thank the referee for this useful suggestion, and we have made clarifying changes to the manuscript as a result. For the emitter quadrature $X(\varphi) = e^{i\varphi}\sigma + e^{-i\varphi}\sigma^\dagger$ we have $\Delta X(\varphi)^2 = 1 - \langle X(\varphi) \rangle^2$. As such, the quadrature $X(\varphi)$ with the least variance is necessarily also that with the greatest expectation value. We can therefore assign any squeezing to be amplitude squeezing, as in phase space the state is narrowed along the same axis as any displacement from the origin. We find that the quadrature with the least variance is that for which the quadrature angle φ is equal to the angle ϕ that represents the phase of the steady-state dipole of the emitter. As we now explain in the paragraphs following Eq. (1) and around Eq. (5), this phase depends on the driving conditions of the emitter, and can in fact be adjusted by varying the phase of the Rabi frequency Ω .

3. When the authors move from the simple atomic case, to the case with the phonon environment, they say they introduce a 'completely generic density operator'. What do they mean by this, and how does this differ from the atomic case?

In the atomic case we explore density operators that are steady-states of the simple spontaneous emission master equation $\dot{\rho} = -i[H_S, \rho] + \Gamma\mathcal{L}_\sigma(\rho)$, which, due to the form of this master equation, do not span the whole Bloch sphere. In the general case, we consider all states within the Bloch sphere. To make this clearer, this point is now explained in more detail, and we give an explicit form for the general density operator we consider in Eq. (4) of the revised manuscript.

4. When phonons are introduced, they only arise in a phonon side-band to the main Lorentzian peak of the two level system of width Γ . Is there no broadening of the line due to these phonon interactions, often termed 'pure dephasing'? This would also affect the saturation parameter s .

While the inclusion of phonons does indeed lead to a phonon sideband in the spectrum, it also gives rise to various T_1 and T_2 like phonon-assisted processes that affect the zero-phonon-line, such as excitation induced dephasing and renormalisation system parameters (e.g. the generalised saturation parameter). All such processes are included in our theory, either in the renormalised system Hamiltonian H_r or within the superoperator $\mathcal{K}_{ph}[\rho(t)]$ in the master equation. Indeed, it is precisely these processes that give rise to regimes of phonon-enhanced squeezing.

To avoid any confusion, in our revised manuscript, when phonons are introduced we now give a more detailed description of their effects, and give an explicit form for the phonon dissipator in the Methods section.

5. In the plots of Fig. 1, the authors take the saturation parameter to a level of 10^7 . Is this feasible for a solid state two level system?

While the driving strengths needed to access the phonon-enhanced squeezing regime are experimentally challenging, they are not beyond the reach of a dedicated effort. We now include a new paragraph discussing precisely this point, which reads:

“We note that although the driving strengths required to reach the regime of vibrationally enhanced squeezing are experimentally challenging, they are not excessively so. Rabi energies of several hundred μeV are fairly routinely achieved experimentally [Unsleber et al., *Optica* 2, 1072 (2015)], and for the realistic parameters used [in this work] the level of squeezing exceeds that possible in the absence of phonons for $\mathcal{S}(\delta) \approx 0.3$ corresponding to a Rabi energy of $\hbar\Omega \approx 0.8$ meV, while the maximum squeezing occurs in the regime $\hbar\Omega \approx 1.5$ meV. Broadly speaking, the vibrationally enhanced regime takes effect when $k_B T < \hbar\sqrt{\Omega^2 + \delta^2}$, and as such lowering the temperature below the 4 K used here would allow it to be observed at lower driving strengths and detuning values.”

6. In Fig. 2 the authors only take the detuning modified saturation parameter S from 0.1 to 10, and the maximum squeezing is around $S = 2$ (the axes would also be less confusing if labelled S and with log ticks on the lower axis of the box). As these are all at a fixed detuning would it not be better to stick to using the saturation parameter s , to allow for easy comparison between figures? Also, where on Fig. 1 corresponds to these plot parameters?

Following the referee’s suggestion, we now label our axes \mathcal{S} and include log ticks in all relevant figures. We choose to use the detuning dependent generalised saturation parameter \mathcal{S} , as it is this parameter which defines the point at which the maximum emission from the emitter is reached off resonance. Thus, any squeezing that happens for $\mathcal{S} < 1$ can truly be said to be below saturation. We believe the distinction between phenomena that occur below or above saturation (defined in this way) is an important one, and as such continue to use \mathcal{S} in the revised manuscript. As the labels above each plot in Fig. 2 indicate, these plots correspond to the lines $\delta = 0$, $\hbar\delta = +1$ meV and $\hbar\delta = -1$ meV in Fig. 1. For the off-resonant cases ($\hbar\delta = \pm 1$ meV), the minimum and maximum scaled driving strengths in Fig. 2 cover the range $s \sim 10^{5.5} - 10^{7.5}$, while in the resonant case $\mathcal{S}(0) = s$ by definition. We have added text to the caption of Fig. 2 of the manuscript to clarify this point.

7. The Wigner functions have clear negativity, which is very interesting. The negativity (not needed for a squeezed state) is much larger than is usually observed for a Fock state of one photon. Is there a limit to the negativity that can be achieved? Could the authors comment on the nonclassicality of their states based on the negativity?

The Wigner functions as presented in the original manuscript were calculated by mapping the state of the quantum dot exciton to the zeroth (vacuum) and first Fock states of a single mode, and therefore take a form which resembles a coherent superposition of these two states. As such, the negativity arises when this superposition contains a sufficient component of the first Fock state, but should not exceed the negativity of this state. The authors wish to apologise, as an error in how the Wigner functions in the original manuscript were normalised led to cases where this latter point was contradicted.

Wigner functions calculated in the way described should be thought of as providing a qualitative representation of the field only. Although the techniques used to generate them have been used before (see *Nature* 525, 222 (2015)), the recent analysis in *Phys. Rev. Lett.* 121, 263603 (2018) suggests the multimode nature of resonance fluorescence fields requires a more careful treatment.

In order to assess the non-classicality of the states explored in our work, we therefore opt not to focus on Wigner functions, but instead analyse the utility of resonance fluorescence states in a phase estimation protocol, where we show that they can provide a sensitivity beyond any classical (coherent) state. This provides a more application-driven insight into the nature of the states produced than the Wigner functions, and is discussed in detail below. The Wigner functions are now presented in the supplementary information.

8. Finally, is the maximum level of squeezing observed useful? Can it be harnessed for any of the applications mentioned in the first paragraph of the manuscript such as QIP applications, sensitive measurements, or QKD systems?

This is an interesting question, which has motivated us to perform a significant new calculation which is now included in the manuscript, with a corresponding new figure. We have analysed the performance of squeezed resonance fluorescence states in a phase estimation protocol, and found that they can indeed provide a greater phase sensitivity than a corresponding classical coherent state with the same number of photons. Moreover, in the regime of vibrationally enhanced squeezing, the generated states give a better phase sensitivity even than the canonical single mode squeezed vacuum state, as is shown in Fig. 3 of the resubmitted manuscript.

As we now state in the discussion section of the manuscript “It is interesting to compare [our] findings with the Fisher information analysis of [Lang and Caves, *Phys. Rev. Lett.* 111, 173601 (2013)], which states that for the setup [considered in our work], the optimal pure single mode input state over all phase estimators is the squeezed vacuum. Our findings therefore suggest that for more general multimode input states the squeezed vacuum ceases to be optimal, or that estimators beyond the difference in output photon numbers must be considered. As such, our results not only illustrate how vibrational environments can give rise to enhanced quadrature squeezing in resonance fluorescence, but also motivate future studies analysing the performance of generalised non-Gaussian multimode states in interferometry.”

Once again we thank the referee for their report and many helpful comments. We have acted upon all points

raised, and believe the manuscript has significantly improved as a result.

Response to Reviewer #2 (Reviewer's comment in italics)

We thank the referee for carefully reviewing our manuscript and for their positive comments. Below we give a detailed response to the referee's questions and suggestions.

The authors use the terminology "Thermal enhancement ..." Of course, phonons are due to thermal oscillations but in fact the way the phonon bath couples to the atom is different than the ordinary thermal field. It is clearly seen in the Hamiltonian (1) of the supplementary material. Therefore, the title of the paper should be changed from "Thermal enhancement ..." to "Phonon enhancement....".

We thank the referee for this suggestion. We agree that the exciton–phonon coupling which leads to the thermally-enhanced squeezing regime is quite different in nature to the usual coupling to the electromagnetic environment. Following the referee's suggestion, we now refer to our central effect as "vibrational enhancement" of quadrature squeezing, while the title has been changed to "Vibrational enhancement of quadrature squeezing and phase sensitivity in resonance fluorescence".

Also the fact that the Walls and Zoller bound can be beaten in the resonance fluorescence of a two-level atom has been known for a very long time. There are papers one by Ficek et al. JOSA B1, 882 (1984) in which it is shown that the Walls and Zoller limit can be beaten in the transient regime of the resonance fluorescence where squeezing twice of that predicted by Walls and Zoller can be achieved in short times. A more general analysis of that prediction was made in 1987 by Aravind, JOSA B4, 1847 (1987).

We thank the referee for bringing these works to our attention. We have added these references and now made it clear in our manuscript that it has previously been identified that the Walls and Zoller limit can be beaten.

We would like to stress, however, that in our analysis with a vibrational environment included, the Walls and Zoller limit is surpassed in the steady-state regime rather than transiently. The effect we identify is therefore not only novel, but (taken together with our new analysis on phase estimation detailed below) also significantly more useful in applications.

The fact that a phonon bath can enhance squeezing is interesting, but I am not convinced how useful it could be for practical applications. The interest is in the generation of squeezed light with significantly reduced fluctuations, I don't see how the fluorescence field with squeezed fluctuations could be applied in practice. Any idea how the fluorescence field could be applied as a source of squeezed light?

We thank the referee for raising this issue, and have performed a significant new analysis as a result. As this was a question also raised by referee #1, we repeat our response here:

We have analysed the performance of squeezed resonance fluorescence states in a phase estimation protocol, and found that they can indeed provide a greater phase sensitivity than a corresponding classical coherent state with the same number of photons. Moreover, in the regime of vibrationally enhanced squeezing, the generated states give a better phase sensitivity than even the canonical single mode squeezed vacuum state, as is shown in Fig. 3 of the resubmitted manuscript.

As we now state in the discussion section of the manuscript "It is interesting to compare [our] findings with the Fisher information analysis of [Lang and Caves, Phys. Rev. Lett. 111, 173601 (2013)], which states that for the setup [considered in our work], the optimal pure single mode input state over all phase estimators is the squeezed vacuum. Our findings therefore suggest that for more general multimode input states the squeezed vacuum ceases to be optimal, or that estimators beyond the difference in output photon numbers must be considered. As such, our results not only illustrate how vibrational environments can give rise to enhanced quadrature squeezing in resonance fluorescence, but also motivate future studies analysing the performance of generalised non-Gaussian multimode states in interferometry."

We would like to thank the referee for their report and many helpful comments. We have acted upon all points raised and believe the manuscript has improved significantly as a result.

Response to Reviewer #3 (Reviewer's comment in italics)

We would like to thank the referee for their detailed report that has led to numerous changes to the manuscript, which we believe has greatly improved as a result. We give below a detailed response to each point raised.

The referee gives a brief summary of our work, before making the following comments:

First and foremost is the introduction does not really motivate what the paper is really about. The first paragraph is about application of squeezed states - generally with many photons. The scheme presented here (at least what the abstract says) is for single photons. What are the applications for squeezed single photons?

To our knowledge there has, until now, been no suggested applications for squeezed single photons. Motivated by this, and the referee's comment, we have performed a significant new analysis to address this. We repeat our response to the other referees here:

We have analysed the performance of squeezed resonance fluorescence states in a phase estimation protocol, and found that they can indeed provide a greater phase sensitivity than a corresponding classical coherent state with the same number of photons. Moreover, in the regime of vibrationally enhanced squeezing, the generated states give a better phase sensitivity than even the canonical single mode squeezed vacuum state, as is shown in Fig. 3 of the resubmitted manuscript.

As we now state in the discussion section of the manuscript "It is interesting to compare [our] findings with the Fisher information analysis of [Lang and Caves, Phys. Rev. Lett. 111, 173601 (2013)], which states that for the setup [considered in our work], the optimal pure single mode input state over all phase estimators is the squeezed vacuum. Our findings therefore suggest that for more general multimode input states the squeezed vacuum ceases to be optimal, or that estimators beyond the difference in output photon numbers must be considered. As such, our results not only illustrate how vibrational environments can give rise to enhanced quadrature squeezing in resonance fluorescence, but also motivate future studies analysing the performance of generalised non-Gaussian multimode states in interferometry."

We have, as a result, significantly altered our manuscript to incorporate these new findings, including making changes to the introduction in order to better motivate our work.

In the introduction it is stated that squeezed states "provide the necessary non-classicality for quantum computing schemes". This is not really true. For CV computation one requires at least a third order nonlinearity. Squeezing is second order.

We thank the referee for pointing out this important detail. We have modified the wording of the introduction accordingly.

The choice of $\hbar = 1$ is confusing especially when the authors consider realistic parameters later in the paper. One should work in SI units and so I recommend the hbars are explicitly put back into the paper. One can see this when $E^{(+)}(t) = E_0(t) - \sqrt{2\Gamma/\pi}\sigma_-$. [The term] E_0 seems from the definition below to be dimensionless while the second terms is $s^{-1/2}$? Notation wise it would be better to use σ_{\pm} rather than sigma σ^{\dagger} . Further the authors themselves refer to ω_l as both an energy and a frequency.

Following the referee's suggestion we have now reintroduced all occurrences of \hbar and used SI units throughout.

Next what is $|X\rangle$ being used for excited state rather than the more traditional $|e\rangle$?

The notation $|X\rangle$ was used to denote the single exciton state in a semiconductor quantum dot. To avoid any confusion, we now use the notation $|e\rangle$ throughout.

On page 2 of the manuscript it is written "we see that the quadrature with the lowest variance is that with $\phi = \psi$. What happens if $\phi = \psi + \pi$ "

With the quadrature defined as $X(\phi) = e^{i\phi}\sigma + e^{-i\phi}\sigma^{\dagger}$ and its variance $\Delta X(\phi)^2 = \langle X(\phi)^2 \rangle - \langle X(\phi) \rangle^2$, we see that $X(\phi + \pi) = -X(\phi)$ while $\Delta X(\phi + \pi) = \Delta X(\phi)$. That is to say, if the quadrature $X(\phi)$ is squeezed, the quadrature $\Delta X(\phi + \pi)$ is also squeezed with the same level of squeezing.

In the revised manuscript, we now discuss the minimum and maximum quadrature variances in greater detail. In particular, we note that the quadrature $X(\varphi = \phi)$ has the minimum variance which is given by $\Delta X(\phi)^2 = 1 - 4|\langle \sigma \rangle|^2$, while the quadrature $X(\varphi = \phi + \pi/2)$ has the greatest variance given by $\Delta X(\phi + \pi/2)^2 = 1$.

Next (as seen in Walls and Milburn), the spectrum is normally defined from -infinity to infinity. What is the reason to define from 0 to infinity in this case?

The referee is correct to point out that in its general form, the expression for the emission spectrum involves an integration over the whole real line. However, for a steady state emission spectrum as considered in our work, the condition $g^{(1)}(-\tau) = g^{(1)}(\tau)^*$ allows the integration to be replaced with one only over positive τ provided the real part is taken, as defined in the manuscript.

It would be useful after eqn (1) to also present the anti squeezed quadrature for completeness. This way we can check that the squeezed state generated is still is a minimum uncertainty state.

As discussed above, we now explicitly show that the quadrature $X(\varphi = \phi)$ has the minimum variance, while the quadrature $X(\varphi = \phi + \pi/2)$ has the greatest variance. Furthermore, in all cases considered we check that the Heisenberg uncertainty relation is obeyed, and discuss whether the states produced are minimum uncertainty states or otherwise.

When the master equation is first introduced on page 2, it is mentioned that "...TLE without additional thermal interactions". The author need to be more explicit that they are working at zero temperature for the master equation presented.

We now explicitly state that the presented master equation is at zero temperature.

With almost any quantum dot system (and in fact solid state system), dephasing is a critical effect and generally more limiting than T_1 . What is its effect here? It probably needs to be included.

To address this point, we would first like to emphasise that the phonon environment included in our model gives rise to both strong dephasing effects and dissipation. In fact, the regime of enhanced squeezing occurs when phonon processes dominate over the spontaneous emission process, with the phonons attempting to thermalise the dot in the laser-driven dressed state basis. As such, the most important dephasing process for quantum dot excitons, caused by coupling to phonons, is indeed included in our model, and is actually one of the causes of the new physics that we reveal.

A related issue is whether our results are robust against other sources of dephasing besides phonons. In fact, since the phonon-enhanced squeezing occurs when the phonon dissipator dominates, our results are qualitatively unaffected by any sources of dephasing that are weak or moderate compared to the spontaneous emission process.

To illustrate this point, in Fig. 1 of this response we show a version of Fig. 2 of the main manuscript, which plots the normally ordered quadrature variance as a function of driving strength. The solid orange curve is our theory including phonons and spontaneous emission but without additional pure-dephasing (identical to the solid orange curve in Fig. 2 of the manuscript), and the dashed red curve shows the effect of also including extra pure-dephasing, here with a rate equal to the spontaneous emission rate. Interestingly, we see that this amount of pure-dephasing is sufficient to completely eliminate the squeezing below saturation that occurs in the absence of phonons (left panel), but the phonon-enhanced squeezing remains above saturation (centre and right panels), with a magnitude that is barely affected. This suggests that the phonon-enhanced squeezing predicted in this work is robust against other sources of dephasing (e.g. charge noise).

We have now including clarifications on these points within the manuscript, as well as a copy of Fig. 1 above within the supplementary information.

Using the symbol S for saturation parameters is problematic as $S(\omega)$ is also used for the spectrum

We now use the symbol $I(\omega)$ for the emission spectrum.

The first paragraph in the second column of page 2 begins "To verify this we consider a TLE described by a completely generic density operator". It would be useful to have a little more explanation on how the results are derived from it.

We thank the referee for this suggestion. We now include an explicit form for this general density operator in Eq. (4) of the revised manuscript, and provide a greater explanation of how it is used.

Next when the thermal reservoir is added, are the steady state just obtained by solving the appropriate master for non zero temperature. Why is it mentioned this is an additional reservoir? Above eqn (3) it is mentioned that ρ_{th} is the TLE thermal state. If it is a thermal state by standard definitions, it should not be squeezed? Please explain

In our model, we first consider a two-level system representing the ground and excited states of an atomic emitter, which is coupled to a reservoir representing the electromagnetic environment into which spontaneous emission occurs (section titled "Squeezing in atomic resonance fluorescence" in the revised manuscript). We then introduce a second reservoir representing some vibrational (phonon) environment to model a solid-state emitter (section titled "Vibrational enhancement of squeezing" in the revised manuscript). In the limit that coupling to the vibrational reservoir dominates over the electromagnetic environment, it will cause the emitter to tend towards a thermal state ρ_{th} , which is determined by the (driven) system Hamiltonian of the emitter and the temperature only. We note, however, that this is a thermal state of the emitter, not of the scattered field. The referee is quite right that a thermal state of the field should not be squeezed, but our results show that a thermal state of the emitter can indeed give rise to a squeezed field, due to the two-level nature of the emitter.

FIG. 1: Normalised quadrature variance calculated using our full phonon model without (orange, solid curves), and with (red, dashed curves) additional non-phonon-induced pure-dephasing.

When we introduce our full quantum dot model (section titled ‘‘Squeezing from a driven quantum dot’’ in the revised manuscript), the steady-state is obtained by solving a master equation, derived from a full microscopic model of the vibrational and electromagnetic environments, which naturally incorporates spontaneous emission and all phonon effects.

These points have been clarified in the revised manuscript.

Next around eqn (3) please show the variance of the other quadrature as well. What is the effect of the thermal noise on it. Do we still have a minimum uncertainty state? Below eqn (3) is stated that the maximum amount of squeezing is : $X(\phi)^2 := 0.25$ is achieved in the low temperature limit. This is a positive value so is it really squeezed?

As discussed above, the quadrature with the maximum variance is $X(\phi + \pi/2)$ and has $\Delta X(\phi + \pi/2)^2 = 1$. The quadratures with the minimum and maximum variances are now given explicitly in the manuscript, and a discussion as to whether the corresponding states are minimal uncertainty states is given. We also now plot the quantum dot purity in the lower panels of Fig. 2 of the revised manuscript, which (as explained in the manuscript) gives us a measure of how close our emitted states are to being minimum uncertainty states.

We apologise, but the value : $X(\phi)^2 := 0.25$ in the original manuscript is a typo, and should read : $X(\phi)^2 := -0.25$. The referee is correct that only negative values of : $X(\phi)^2$: correspond to squeezed states.

The model used here is showing not spatial character for the phonon bath but we have a 3D system. Generally when there is not spatial confinement, we need the e^{ikx} term. However as there is no cavity etc, this may be important. Please comment on it?

The spatial character of the quantum dot and its coupling to the phonon environment is included in our treatment. It is contained within the spectral density as a cut-off frequency ν_c , which is inversely proportional to the spatial extent of the quantum dot exciton. In our calculations, we use a realistic value of $\hbar\nu_c = 1.45$ meV which is consistent with experimental results in Ramsey *et al.*, PRL 105, 177402 (2010). We now make this point clear in the Methods section.

The electron–phonon interaction is given by the Hamiltonian H_{PH} . However this makes it look like a coherent interaction? Is it? The g_k will vary over time in terms of both amplitude and phase. Should we not have written the last part as $g_k b_k^\dagger + g_k^ b_k$?*

The electron–phonon coupling is modelled in a rigorous manner, being as it is included in the microscopic Hamiltonian. At first sight it may appear coherent, but in fact the (continuous) multimode nature of the phonon environment means that (excluding very short times) the interaction is not coherent, and gives rise to various effects of both dissipation and dephasing in nature, captured in the phonon dissipator K_{ph} . The electron–phonon coupling term is written in the Schrödinger picture, and is therefore time-independent, allowing us to take g_k real without loss of generality.

The dissipator K_{ph} is not really described in the main text of the paper, nor what form it has. It is in the supp material but a description needs to be in the main text.

We thank the referee for this suggestion. The full form of the dissipator is now given in the Methods section of the main text, while its action and effects are discussed in more detail when it is first introduced.

When the Wigner function is described I have a problem with it. The emission of the light field is not into a

single mode and in fact is radiated into free space (3D). How is this accounted for?

Wigner functions calculated in the way described should be thought of as providing a qualitative representation of the field only. Although the techniques used to generate them have been used before (see Nature 525, 222 (2015)), the recent analysis in Phys. Rev. Lett. 121, 263603 (2018) suggests the multimode nature of resonance fluorescence fields requires a more careful treatment, as the referee alludes to. In our revised manuscript, we therefore opt not to focus on Wigner functions, and instead present them only in the supplementary information.

Next it is well known that the Wigner function for the state of an atom can be negative, even if it is in its ground or excited state. I am also confused that the Wigner function for the squeezed state is negative. For optical fields it is strictly positive as it comes from a Gaussian state. A significant discussion needs to be made around this point.

The optical fields we explore in our work are produced by a two-level emitter. Since the emitter cannot be described as an oscillator, notions of Gaussianity do not apply to it, and by extension, it can produce optical fields that need not be Gaussian, nor have positive Wigner functions. The canonical example being the generation of single photons from an emitter, which necessarily has a non-Gaussian and negative Wigner function. Nevertheless, as mentioned above, in our revised manuscript we opt to focus instead on the utility of states produced in resonance fluorescence in phase estimation protocols, which we believe gives a more application-driven and accurate characterisation of their non-classicality. The Wigner functions are now presented in the supplementary information.

Finally to show that the states generated are useful, how about illustrating their application in a metrology protocol.

As mentioned above, we now include a major new analysis, which shows that squeezed states generated in resonance fluorescence are not only useful in phase estimation protocols, but can even outperform squeezed vacuum input states in phase sensitivity. This result is a significant finding in its own right, and we thank the referee for providing us with the motivation to perform this extra study.

Now as a more minor comments, the format of this paper is not that used for Nature Communications.

The manuscript has now been formatted as a Nature Communications article.

To summarize, I believe this is an interesting article, but it is not clear to me currently whether it would be of interest to the wider community. The shortened nature of the manuscript means important details are missing (which could be included given the non letter format of Nat Comm).

We thank the referee for their positive comments and insightful criticism. With the extensive changes to the manuscript and the addition of significant new results and analysis, we believe that we have fully addressed all the points raised. Furthermore, we believe that the phase estimation scheme we have now introduced into the manuscript greatly broadens the appeal of the work.

List of Changes

- Following the suggestion of referee 2, and to reflect the significant new analysis included in the manuscript, we have changed its title to “Vibrational enhancement of quadrature squeezing and phase sensitivity in resonance fluorescence”.
- The format of the manuscript has been changed to that of Nature Comms., including section headings and a Methods section.
- Following comments made by all referees, we have significantly restructured the introduction to better put our results in context.
- In response to comments made by referees 1 and 2, when the quadratures are first introduced around Eq. (1), and after Eq. (3), we now discuss the minimum and maximum variance quadratures, and the conditions under which a state is a minimum uncertainty state. We now also give a discussion concerning which is the squeezed quadrature, and how it can be adjusted.
- Following referee 3’s suggestion, we now use the symbol $I(\omega)$ for the emission spectrum.
- In response to referee 1, we now give an explicit form for the general density operator we consider in Eq. (4).
- Following a comment made by referee 1, all relevant figures are labelled s or S with log ticks shown.
- We have added text to the caption of Fig 2 to aid comparison between figures.
- In order to assess whether resonance fluorescence states are minimum uncertainty states, we have modified Fig. 2 to include plots showing the purity of the quantum dot exciton.
- In response to comments made by referee #1, we now include a paragraph that discusses the driving strengths needed to reach the phonon-enhanced squeezing regime.
- Fig. 3 of the original manuscript, and its discussion have been moved to the supplementary information.
- Fig. 3 of the revised manuscript, and the whole of the section titled “Resonance fluorescence and phase estimation” are new, and detail the results of our new analysis.
- Following a comment made by referee #3, the Methods section now contains a full definition of the phonon dissipator K_{ph} .
- Reference added to Vahlbruch, Phys. Rev. Lett. 117, 110801 (2018)
- Reference added to Lang and Caves, Phys. Rev. Lett. 111, 123601 (2013)
- Reference added to Marek, Jeong and Kim, Phys. Rev. A 78, 063811 (2008)
- Reference added to Aravind, J. Opt. Soc. Am. B 4, 1847 (1987)
- Reference added to Ficek, Tanaś and Kielich, J. Opt. Soc. Am. B 1, 882 (1984)
- Reference added to Ficek and Swain, J. Opt. Soc. Am. B 14, 258 (1997)
- Reference added to Unsleber et al., Optica 2, 1072 (2015).
- Numerous small changes have been made to the text to improve readability.

REVIEWERS' COMMENTS:

Reviewer #1 (Remarks to the Author):

Iles-Smith et al. have made significant changes to the originally submitted manuscript, including a large portion of new work to demonstrate the application of the generated squeezed resonance fluorescence state to phase estimation. The authors have addressed all of my questions satisfactorily, and I believe this manuscript to be of high interest to the readers of Nature Communications.

I have a few minor queries and comments:

1. Top of page 2, column 2, I_{inc} should be $I_{\text{inc}}(\omega)$.
2. In the color bar of figure 1 (c) and (d), the dashed line should be at 0.125, but looks to be at 0.14?
3. The region covered by Fig. 2(a) is a very small part of Fig. 1 (a) and (c). The largest coherent fraction on resonance is at higher driving strengths, with s between 10^5 and 10^7 . Might it be interesting to show the evolution of $\langle \Delta X(\phi)^2 \rangle$ and I for a larger range on resonance in the supplementary information?
4. There is a rather complicated evolution of the squeezed vacuum input state in the negative detuning case of Fig. 3(c). Is this due to the change in the number of photons generated in this regime, which changes the normalization? It also looks here as though the case without phonons achieves a better sensitivity than the squeezed vacuum case. A little more discussion of this final plot would be appreciated.
5. In equations 10-12 there is no Δ_{yy} term, but it is defined after. Should the Δ_{xx} in Eq. 10 be $\Delta_{\alpha\alpha}$? Similarly, is there a need for ζ_{α} to be general, or could only ζ_z be defined? I also think the inline equations that follow Eq. 12 might be more legible if separated onto new lines.
6. In the supplementary the word "FILED" should be "FIELD".

Reviewer #2 (Remarks to the Author):

The authors have considered my comments and suggestions and have incorporated all of them in the revised version of the manuscript. I find the revised version suitable for publication.

Reviewer #3 (Remarks to the Author):

I believe the authors have addressed the points I raised in my previous round of review in a nice way. The paper has been significantly improved and the refocused paper (including new title and introduction) will be a valuable addition to the community and literature.

A minor technical point is that the anti commutator $\{ \}$ notation in the emission dissipator $L(\rho)$ formula has not been defined.

Response to Reviewer #1 (Referee's comment in italics)

1. *Top of page 2, column 2, I_{inc} should be $I_{inc}(\omega)$.*

We thank the referee for pointing out this omission. The manuscript has been updated accordingly.

2. *In the color bar of figure 1 (c) and (d), the dashed line should be at 0.125, but looks to be at 0.14?*

The referee is correct that the dashed line should correspond to 0.125. This error has been corrected.

3. *The region covered by Fig. 2(a) is a very small part of Fig. 1 (a) and (c). The largest coherent fraction on resonance is at higher driving strengths, with s between 10^5 and 10^7 . Might it be interesting to show the evolution of $:\Delta X(\phi)^2$: and l for a larger range on resonance in the supplementary information?*

Following the referee's suggestion, we have now added a section and figure (Note 3) to the Supplementary information which shows the quadrature variance over the full range of driving strengths.

4. *There is a rather complicated evolution of the squeezed vacuum input state in the negative detuning case of Fig. 3(c). Is this due to the change in the number of photons generated in this regime, which changes the normalisation? It also looks here as though the case without phonons achieves a better sensitivity than the squeezed vacuum case. A little more discussion of this final plot would be appreciated.*

For the parameters used in the manuscript and those varied in Fig. 3, the dominant change in the phase sensitivity figure of merit for the squeezed vacuum arises from the change in photon number and corresponding normalisation factor, as the referee suggests. The referee is also correct to point out that in the negative detuning case the resonance fluorescence field can outperform the squeezed vacuum, even in the absence of phonons.

While this is true and was not clearly reflected in the previous version of our manuscript, we point out that the minimum of our figure of merit in the squeezed vacuum case is $\mathcal{F} = 0.5$ for the parameters used in the manuscript. While the resonance fluorescence field can outperform the squeezed vacuum in the negative detuning case even without phonons for the same photon number, it does not reach values less than $\mathcal{F} = 0.5$. Only in the resonance fluorescence case for positive detunings, including phonons, can \mathcal{F} reach values less than 0.5. As such, it is accurate to say that the resonance fluorescence field only outperforms the best squeezed vacuum figure of merit when phonons are included.

In response to these identifications, we have slightly modified the relevant figure to include the squeezed vacuum minimum of $\mathcal{F} = 0.5$ as a gridline. We have also changed the wording when this figure is described and in our discussion, and added additional details following Eq. (7).

5. *In equations 10-12 there is no Δ_{yy} term, but it is defined after. Should the Δ_{xx} in Eq. 10 be $\Delta_{\alpha\alpha}$? Similarly, is there a need for ζ_{α} to be general, or could only ζ_z be defined? I also think the inline equations that follow Eq. 12 might be more legible if separated onto new lines.*

The referee's queries here arise from a small number of typos present in the revised manuscript. These have now been corrected in the current second revision, and a number of the following inline equations have been made into display equations.

6. *In the supplementary the word "FILED" should be "FIELD".*

This typo has been corrected.

Response to Reviewer #3 (Referee's comment in italics)

A minor technical point is that the anti commutator $\{\}$ notation in the emission dissipator $L(\rho)$ formula has not been defined.

The anticommutator is now defined where it arises in the Supplementary information.

List of Changes

- Abstract has been rewritten in order to better motivate our work.
- Reference added to PRA 88, 023837 (2013). Ref. [31]
- Reference added to PRL 109, 013601 (2012). Ref. [32]
- Titles added to all Figures.
- Definition of the spectral density $J(\omega)$ moved from methods into main text.
- Derivation of phase estimation results moved from Supplementary Information into Methods in the main text.
- Notation for Eqs. (9) to (12) corrected in response to referee one's comments.
- Fig. 3 labels changed to comply with standards.
- Gridline added to Fig. 3 (b–d) to show the figure of merit minimum for the squeezed vacuum.
- Discussion when referencing Fig. 3 and in the discussion modified following comments from referee one.
- Data availability statement added.
- Author contributions statement added.
- Competing Financial Interests statement added.
- Section titles in Supplementary Information changed to Supplementary Note One etc.
- Minor changes made to notation throughout in order to comply with Nature Comms. standards.
- Titles and page numbers added to all references in the main text and Supplement.